# Refined Partitioning Boosts MGDA: Introducing RP-MGDA for Multi-Objective Learning

## Abstract

Multi-objective optimization is a critical topic in the field of machine learning, with applications in multi-task learning, federated learning, reinforcement learning, and more. In this paper, to elevate the performance of the widely used Multiple Gradient Descent Algorithm (MGDA) in multi-objective optimization tasks, we introduce a novel version of MGDA through Refined Partitioning (RP-MGDA). RP-MGDA leverages the concept of 'refined partitioning', where variables are strategically partitioned and grouped in order to improve the optimization process, in contrast to vanilla MGDA which ignores potential variable structure and, as a result, treats all parameters as one variable. Our examples and experiments showcase the effectiveness of RP-MGDA compared to MGDA under various scenarios. We provide insights into the underlying mechanisms of RP-MGDA and demonstrate its potential applications. Specifically, the concept of refined variable partitioning in RP-MGDA is not limited solely to MGDA and holds promise for enhancing other multi-objective gradient methods (e.g., PCGrad, CAGrad).

## 1 Introduction

Multi-objective optimization deals with problems that involve optimizing multiple objective functions simultaneously over shared parameters and aims to find non-inferior solutions. It is very common in machine learning to address multiple aspects concurrently and perform optimization. For example, one may need to consider both the accuracy and fairness of a model simultaneously in fair ML, or one may need to optimize the performances of multiple entities, as seen in federated learning and reinforcement learning. The optimal solutions of multi-objective optimization problems are called 'Pareto optimal' solutions

Among different approaches to solve multi-objective optimization problems, one method has become quite popular recently is the Multiple Gradient Descent Algorithm (MGDA) Mukai (1980); Fliege & Svaiter (2000); Désidéri (2012). MGDA is an extension of classical gradient descent algorithm to the multi-objective setting. Compared to other commonly used approaches that handle multiple objectives in machine learning (e.g. weighted sum approach), MGDA has the unique feature of not requiring predetermined scalarization of the objective and can directly optimize the vector-valued multi-objective optimization problem. Practically, MGDA has been recently applied to many deep learning areas including multitask learning Sener & Koltun (2018); Lin et al. (2019), federated learning Hu et al. (2022), generative model Albuquerque et al. (2019) and etc. MGDA has also attracted theoretical attention, new properties and variants of the method has been studied since then (e.g.,Mercier et al. (2018); O. Montonen & Mäkelä (2018); Tanabe et al. (2019); Assunção et al. (2021)).

In a nutshell, each iteration, MGDA finds a common descent direction to all objectives and descend along it with a proper step size. This makes MGDA a descent method for multi-objective problems, unlike approaches such as weighted sum which do not have this descending property for every objective. Under mild assumptions, MGDA is proven to converge to Pareto stationarity, which is a necessary condition for Pareto optimality, analogous to how traditional gradient descent converges to stationary solutions in the single objective case.

However, unlike in the single objective setting where stationary solution must be global optimal when objective is convex, *Pareto stationary* solution is only weakly Pareto optimal when objectives

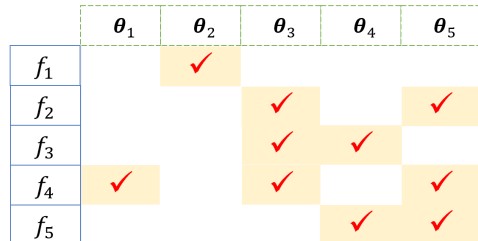

(a) Variable dependency structure of the multi-objective optimization problem

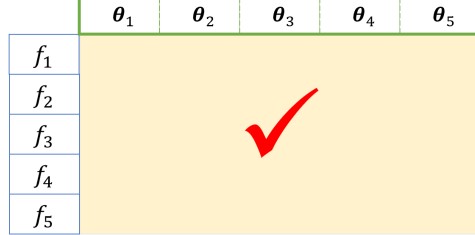

(b) MGDA: groups everything together

(c) RP-MGDA: strategically partitions the variables

Figure 1: A simplified illustration of the difference between RP-MGDA and vanilla MGDA.

are convex. This results in MGDA converging to inferior solutions even for some benign convex problems and thus opens up possibilities for improvement of the method.

In this paper, we identify an important multi-objective scenario where the objective dependencies on variables are structured, and demonstrate that straightforward application of MGDA to these problems can lead to inferior solutions which can be significantly improved if we take the variable structure into account and do careful partitioning. To this end, we propose RP-MGDA ('RP' stands for 'refined partitioning') which applies MGDA on elaborately partitioned subsets of variables, that is, roughly speaking, at least as good as MGDA and can strictly improve over MGDA for problems with specific variable dependency structures.

We give motivating examples to demonstrate how and why MGDA fails on these problems, and propose a systematic approach to do refined partitioning with the inspiration from the examples. We also illustrate through experiments the properties and effectiveness of RP-MGDA.

We summarize out contribution as follows:

- In §3.2 and §A.2, we give motivating examples on how MGDA fails on certain convex (and even individually strictly convex objective) problems and make important observations. These examples reveal an inherent drawback of MGDA that needs to be fixed only by taking variable structure into account.

- In §4, motivated from the previous examples, we propose a systematic approach to do refined variable partitioning with MGDA, resulting in the novel RP-MGDA algorithm. We give theoretic analysis of the algorithm, and demonstrate its superiority over MGDA.

- In §5, we perform various experiments, covering different aspects, to analyze the effectiveness of RP-MGDA in comparison to MGDA.

- The idea of refined variable partitioning can potentially benefit other MOO methods.

## 2 BACKGROUND

In this section, we briefly review the concepts of Pareto optimality and Pareto stationarity in multi-objective optimization. We then introduce the widely used Multiple Gradient Descent Algorithm.

## 2.1 MULTI-OBJECTIVE OPTIMIZATION (MOO)

In mathematical terms, a Multi-Objective Optimization (MOO) problem can be written as

$$\min_{\mathbf{w} \in \mathbb{R}^d} \mathbf{f}(\mathbf{w}) := (f_1(\mathbf{w}), f_2(\mathbf{w}), \ldots, f_m(\mathbf{w})), \tag{1}$$

where the minimum is defined wrt the *partial* ordering[1] :

$$\mathbf{f}(\mathbf{w}) \leq \mathbf{f}(\mathbf{z}) \iff \forall i = 1, \ldots, m, \ f_i(\mathbf{w}) \leq f_i(\mathbf{z}). \tag{2}$$

Unlike single objective optimization, with multiple objectives it is possible that

$$\mathbf{f}(\mathbf{w}) \nleq \mathbf{f}(\mathbf{z}) \text{ and } \mathbf{f}(\mathbf{z}) \nleq \mathbf{f}(\mathbf{w}), \tag{3}$$

in which case we say $\mathbf{w}$ and $\mathbf{z}$ are not comparable. As a result, there is usually a set of solutions that are all optimal (aka. *Pareto Optimal*) for a given MOO problem, which is called *Pareto front*.

## 2.2 PARETO OPTIMALITY AND PARETO STATIONARITY

**Definition 1** (Pareto Optimality). *We call* $\mathbf{w}_*$ *a* Pareto optimal *solution of* (1) *if its objective value* $\mathbf{f}(\mathbf{w}_*)$ *is a minimum element w.r.t. the partial ordering in* (2); *equivalently,* $\forall \mathbf{w}, \ \mathbf{f}(\mathbf{w}) \leq \mathbf{f}(\mathbf{w}_*)$ *implies* $\mathbf{f}(\mathbf{w}) = \mathbf{f}(\mathbf{w}_*)$[2]. *In other words, it is not possible to improve* any *component objective in* $\mathbf{f}(\mathbf{w}_*)$ *without compromising* some *other objective.*

**Definition 2** (Weakly Pareto Optimality). *We call* $\mathbf{w}_*$ *a* weakly *Pareto optimal solution if there does not exist any* $\mathbf{w}$ *such that* $\mathbf{f}(\mathbf{w}) < \mathbf{f}(\mathbf{w}_*)$, *i.e., it is not possible to improve* all *component objectives in* $\mathbf{f}(\mathbf{w}_*)$ *simultaneously.*

Clearly, any Pareto optimal solution is also weakly Pareto optimal but the converse may not hold.

Next, we introduce the concept of *Pareto Sationarity* (also referred to as *Pareto Criticality*) which is the first order necessary condition for Pareto optimality.

**Definition 3** (Pareto Stationarity). *We call* $\mathbf{w}_*$ *a Pareto stationary solution iff* some *convex combination of the gradients* $\{\nabla f_i(\mathbf{w}^*)\}$ *vanishes,* i.e. *there exists some* $\boldsymbol{\lambda} \geq 0$ *such that* $\sum_i \lambda_i = 1$ *and* $\sum_i \lambda_i \nabla f_i(\mathbf{w}^*) = \mathbf{0}$.

The following lemma states the relation between the above three concepts.

**Lemma 1** (Mukai (1980)). *Any Pareto optimal solution is Pareto stationary. Conversely, if all functions are convex, then any Pareto stationary solution is weakly Pareto optimal; if all* $f_i$ *are strictly convex, then any Pareto stationary solution is Pareto optimal.*

## 2.3 MULTIPLE GRADIENT DESCENT ALGORITHM (MGDA)

MGDA has been independently proposed in Mukai (1980); Fliege & Svaiter (2000) and Désidéri (2012) as a gradient-based method to find Pareto stationary solution of an MOO problem and has gained notable attention in the machine learning field in recent years, largely because of its gradient-based nature, in contrast to traditional MOO techniques.

In each iteration, MGDA seeks a new solution that minimizes the maximum change among all objectives, i.e.,

$$\tilde{\mathbf{w}}^{t+1} = \operatorname*{argmin}_{\mathbf{w}} \ \max_{\boldsymbol{\lambda} \in \Delta} \ \boldsymbol{\lambda}^\top (\mathbf{f}(\mathbf{w}) - \mathbf{f}(\tilde{\mathbf{w}}^t)), \tag{4}$$

With first order approximation and quadratic upper bound on $\mathbf{f}(\mathbf{w})$, we obtain the dual problem:

$$\max_{\boldsymbol{\lambda} \in \Delta} \ \min_{\mathbf{w}} \ \boldsymbol{\lambda}^\top J_{\mathbf{f}}^\top(\mathbf{w}^t)(\mathbf{w} - \mathbf{w}^t) + \tfrac{1}{2\eta} \|\mathbf{w} - \mathbf{w}^t\|^2. \tag{5}$$

where $J_{\mathbf{f}} = [\nabla f_1, \ldots, \nabla f_m] \in \mathbb{R}^{d \times m}$ is the Jacobian and $\eta > 0$ is the step size.

---

[1]We remind that algebraic operations such as $\leq$ and $+$, when applied to a vector with another vector *or scalar*, are always performed component-wise

[2]We focus on minimization in this work, while maximization can be treated similarly

Solving for $\mathbf{w}$ by setting its derivative to $\mathbf{0}$, we deduce the following 'min-norm' procedure:

$$\mathbf{d}^t = J_{\mathbf{f}}(\mathbf{w}^t)\boldsymbol{\lambda}_*^t, \quad \text{where} \quad \boldsymbol{\lambda}_*^t = \underset{\boldsymbol{\lambda} \in \Delta}{\operatorname{argmin}} \ \|J_{\mathbf{f}}(\mathbf{w}^t)\boldsymbol{\lambda}\|^2. \tag{6}$$

and then perform the descent along the direction $\mathbf{d}^t$:

$$\mathbf{w}^{t+1} = \mathbf{w}^t - \eta\mathbf{d}^t. \tag{7}$$

Executing iterations (6) and (7) repeatedly gives the Multiple Gradient Descent Algorithm (MGDA). Common stopping criteria include $\|\mathbf{d}^t\| = 0$ (or $<$ threshold $\epsilon$) or after fixed number of iterations.

It can be shown that MGDA decreases all objectives simultaneously in every iteration and either stops at or converges to a Pareto stationary solution in the end (Fliege et al., 2019).

## 3 LIMITATIONS OF MGDA AND REMEDIATION

In this section, we first review some related works that address specific limitations of MGDA, offering proposed solutions to enhance its performance. Then, we shed light on an often overlooked aspect of MGDA—the limitation of Pareto stationarity—and provide an illustrative example that explores the issue, including what develops into our novel RP-MGDA algorithm in §4.

### 3.1 RELATED WORKS

Initially, MGDA was introduced and analyzed with the presumption that exact gradients of objective functions could be efficiently computed. Nevertheless, when MGDA is adapted for deep learning applications, it encounters the challenge of accommodating stochasticity in mini-batch gradients, where exact gradients are not always practical. This circumstance can lead to biased descent directions, impeding the convergence process. Recent research efforts, exemplified by works like Zhou et al. (2022) and Fernando et al. (2023), have rigorously examined this issue and put forth variants aimed at ameliorating it. Concurrently, another line of research, including Assunção et al. (2021), has extended the idea of MGDA to encompass the realm of constrained multi-objective optimization problems and problems with other special properties.

Note that even under the assumptions of exact gradients and unconstrained problems, applying MGDA can lead to the convergence to undesired stationary solutions, as we will see below.

### 3.2 THE PROBLEM WITH (LACKING) STRICT CONVEXITY

Based on Lemma 1, it is established that MGDA converges to Pareto stationary solutions, ensuring Pareto optimality when all objectives are strictly convex. Nevertheless, the subsequent toy example may initially appear to challenge this otherwise robust guarantee.

**An illustrative example.** Consider the bi-objective minimization problem:

$$\min_{\mathbf{w}} \ (f_1(\mathbf{w}), f_2(\mathbf{w})),$$
$$f_1(\mathbf{w}) = (w_1 - 1)^2, f_2(\mathbf{w}) = w_2^2. \tag{8}$$

where $\mathbf{w} := (w_1, w_2)$. Here, note that both $f_1$ and $f_2$ are strictly convex w.r.t the sole parameter they depend on, i.e. $w_1$ and $w_2$ respectively.

Although the problem is presented as bi-objective optimization with shared variables $(w_1, w_2)$, it is apparent that both objectives can be separately optimized, as $f_1$ solely relies on $w_1$ and $f_2$ solely relies on $w_2$. If we do so, the optimal solution we get is $\mathbf{w} = (1, 0)$ with corresponding objective function values being $(f_1, f_2) = (0, 0)$.

However, if we try to apply MGDA directly to this problem, for example, with $\eta = 0.1$ and different initializations $\mathbf{w}^0 = (2, 2), (0, 0.01)$ and $(0.9, 0.2)$, the algorithm converges to sub-optimal solutions with function values $(f_1, f_2) = (0, 2.92), (0.9999, 0)$ and $(0, 0.0292)$ respectively, all of which are dominated by the Pareto optimal $(f_1, f_2) = (0, 0)$. Indeed, it can be easily verified that $\{\mathbf{w} = (a, 0) \cup (1, b)\}$ are all Pareto stationary solutions for this problem and, meanwhile, sub-optimal whenever $a \neq 1, b \neq 0$, which seems to contradict the nice guarantee of Lemma 1.

**Remark 1** (Individual strict convexity is not enough). *In the above example, $f_1$ is strictly convex w.r.t. $w_1$ and $f_2$ is strictly convex w.r.t. $w_2$, but neither of them is strictly convex w.r.t $\mathbf{w}$ (inspect the Hessians). This becomes an issue since vanilla MGDA does not factor in variable structure and treats $\mathbf{w}$ holistically.*

**Remark 2** (Joint strict convexity is not always possible). *We argue that within specific variable dependency structures, e.g., having degenerate Hessians, joint strict convexity is an unreasonable assumption, and thus sub-optimal Pareto stationary solutions are inevitable when employing MGDA to these problems. In appendix Appendix A.2, we analyze in depth a personalized federated learning setting where joint strict convexity cannot hold, leading to the failure of MGDA in a similar manner.*

We see from the above example that partitioning the variables and then separately optimizing them is a simple but effective approach. However, generalizing the idea for more complicated problems to get appropriate partitioning of variables is not straightforward, as we will discuss shortly.

## 4 MULTIPLE GRADIENT DESCENT ALGORITHM WITH REFINED PARTITIONING

In this section, we deal with MOO problems with general variable dependency structures and introduce Multiple Gradient Descent Algorithm with Refined Partitioning (RP-MGDA) that gives a systematic approach to the grouping and partitioning of variables. Our goal is to partition the variables into refined groups while still ensuring that the application of an MGDA-like algorithm to these groups *separately* remains intact.

### 4.1 WHY NOT PARTITION EVERYTHING?

The initial question that may arise is why not partition every variable and apply MGDA individually, potentially resulting in a coordinate-wise MGDA in the extreme case. The answer lies in the fact that such an approach can prematurely terminate before achieving Pareto stationary solutions, effectively disrupting the MGDA algorithm. We show a typical example below to illustrate this.

Consider a bi-objective optimization problem in $\mathbb{R}^2$. Where the two gradients at some iteration are, for example, $\nabla f_1 = (1, 3)$ and $\nabla f_2 = (-3, -1)$. According to MGDA, there is a common descent direction along $-\mathbf{d} = (1, -1)$. However, if we partition the two coordinates and apply MGDA separately, then due to opposite gradient signs of each coordinate, the algorithm will stop immediately (since the min-norm element in the convex hull is zero for both coordinate, see Definition 3).

In general, when dealing with dense variable structures, aggressive partitioning is not feasible for the reason outlined above (more details in the next subsection).

### 4.2 PROCEDURE OF REFINED PARTITIONING

Optimizing separately over partitioned variables can reach superior solutions than MGDA (e.g. §3.2 and Appendix A.2. On the other hand, we should be careful not to partition too aggressively, which may cause the algorithm to fail detrimentally (§4.1). To strike a balance between partitioning and merging, we propose the following procedure for refined partitioning, which consists of three rules.

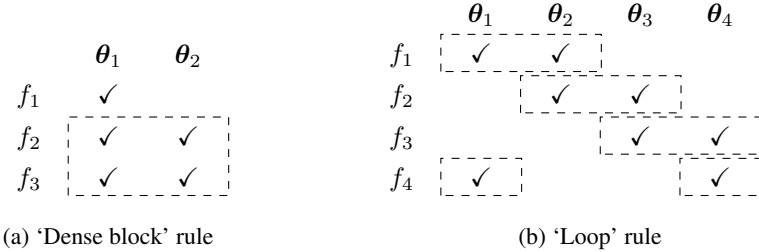

(a) 'Dense block' rule            (b) 'Loop' rule

Figure 2: Illustration of the 'Dense block' and 'Loop' rule.

---

**Algorithm 1:** Multiple Gradient Descent Algorithm with Refined Partitioning (RP-MGDA)

---

**Input:** Objectives-Variables dependency structure $M$, initial model weights
$\mathbf{W}^0 = (\mathbf{w}_1^0, \ldots, \mathbf{w}_n^0)$, learning rate $\eta$.

1 Partition the weight variables $\mathbf{W}$ according to procedure (4.2.1) based on $M$:
$(\mathbf{v}_1, \ldots, \mathbf{v}_l) = \text{REFINED\_PARTITION}(M)$

2 Initialize $(\mathbf{v}_1^0, \ldots, \mathbf{v}_l^0)$ according to $\mathbf{W}^0$

3 **for** $t = 0, 1, 2, \ldots, T-1$ **do**

   (In parallel)

4   $\mathbf{v}_1^{t+1} = \mathbf{v}_1^t - \eta \mathbf{d}_1^t$

5   $\mathbf{v}_2^{t+1} = \mathbf{v}_2^t - \eta \mathbf{d}_2^t$

   $\ldots$

6   $\mathbf{v}_l^{t+1} = \mathbf{v}_l^t - \eta \mathbf{d}_l^t$,   where $\mathbf{d}_i^t$ are solved from min-norm oracle (6)

**Output:** Final model weights $\mathbf{W}^T = (\mathbf{v}_1^T, \mathbf{v}_2^T, \ldots, \mathbf{v}_l^T)$

---

**Definition of key terms**   If $f_i$ depends on $\boldsymbol{\theta}_k$ and $f_j$ depends on $\boldsymbol{\theta}_k$, we say $f_i$ and $f_j$ has *shared variable* $\boldsymbol{\theta}_k$; an *edge* of color $i$ exists between 2 nodes (e.g. $\boldsymbol{\theta}_k$ and $\boldsymbol{\theta}_l$) iff $\exists\, f_i$ that depends on both $\boldsymbol{\theta}_k$ and $\boldsymbol{\theta}_l$; when variables are *merged*, we group the variables together and treat it holistically as a new variable, and if $f_i$ previously depends on any of the variable within the group, it now depends on the new merged variable.

### 4.2.1   REFINED PARTITIONING PROCEDURE

**Rule (I) ['Dense block' Merge]**

If $\exists\, f_i, f_j$ with $\geq 2$ shared variables, then these shared variables are merged.

**Rule (II) ['Loop' Merge]** If a loop of length $\geq 3$ with *different* edge colors exists in the variable structure, then all variables in that loop must be merged.

**Rule (III) [Partitioning]** After exhaustively applying rule (i) and (ii) until no variables can be merged, the remaining variables are partitioned.

So if the input variable dependency structure has $n$ variables $(\mathbf{w}_1, \ldots, \mathbf{w}_n)$, the output will be $l$ (merged) variables $(\mathbf{v}_1, \ldots, \mathbf{v}_l)$, where obviously $l \leq n$.

**Remark 3** (RP-MGDA to MGDA)**.** *In the extreme case where the variable structure is super dense (e.g., $\exists$ loop of length $n$) and all variables are merged together, the output of the procedure is $1$ merged variable $\mathbf{v}_1 = \mathbf{W}$, and as a result RP-MGDA will reduce to the familiar MGDA.*

### 4.3   ALGORITHM AND ANALYSIS

Following the above refined partitioning procedure, we present the full algorithm of RP-MGDA in Algorithm 1 in pseudo-code. The algorithm terminates when $\sum_i \|\mathbf{d}_i^t\|^2 = 0$. Alternatively, $\sum_i \|\mathbf{d}_i^t\|^2 < \epsilon$, if looking for an approximate stationary solution.

Building upon Remark 3, it is evident that RP-MGDA can be reduced to MGDA, which already ensures a baseline performance. The subsequent theorem demonstrates that RP-MGDA also maintains a Pareto stationarity guarantee in non-degenerating scenarios. Proof relegated to Appendix A.1.

**Theorem 1.** *RP-MGDA must attain a Pareto stationary solution upon termination.*

The following theorem provides a theoretical perspective on the superiority of RP-MGDA.

**Theorem 2.** *There exists a non-trivial[3] convex multi-objective optimization problem where the solutions of RP-MGDA are provably Pareto optimal while MGDA is not, under the same assumptions.*

The proof of Theorem 2 is relegated to Appendix A.2, where it is supported by detailed analysis of the personalized federated learning scenario.

---

[3]Here non-trivial means the problem has a variable structure such that when RP-MGDA is applied, it does not degrade to single objective gradient descents on each variable separately

## 5 EXPERIMENTS

In this section, we present experimental results comparing RP-MGDA with MGDA across varying variable dependency structures and convexity assumptions. Notably, we observe significant differences in solution quality even for elementary examples. In §5.1, we illustrate the Pareto (stationary) fronts of RP-MGDA and MGDA in both strictly and non-strictly convex bi-objective optimization problems. This empirical analysis highlights the advantages of RP-MGDA, and we also demonstrate that MGDA's limitations in this context cannot be overcome by adding regularization to induce strong convexity, which is a common technique in optimization. In §5.2, we apply the RP-MGDA rules developed in §4.2 to a 5-by-5 randomly generated dependency matrix example. Here, we compare RP-MGDA's performance to that of MGDA and showcase the superior results by RP-MGDA.

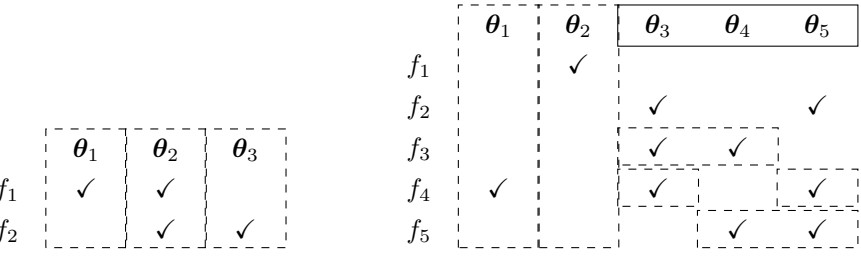

Figure 3: Variable dependency structures in §5.1 and §5.2.

### 5.1 CHAIN EXAMPLE AND PARETO FRONT

We consider the quadratic setup: $f_1(\boldsymbol{\theta}) = \theta_1^2 + \theta_2^2$, $f_2(\boldsymbol{\theta}) = (\theta_2 - 1)^2 + (\theta_3 - 1)^2$. With grid search of different initializations, we draw the Pareto (stationary) fronts of MGDA and RP-MGDA respectively. Result is shown in Figure 4.

We can see clearly that MGDA can arrive at non-optimal Pareto stationary solutions (i.e. those with $f_1 = 0, f_2 > 1$ and $f_1 > 1, f_2 = 0$), while RP-MGDA will always reach solutions on Pareto front. Moreover, if we initialize the weights of RP-MGDA with solutions reached by MGDA, then RP-MGDA will leave the optimal solutions unchanged and bring the non-optimal stationary solutions of MGDA to the end points on the Pareto front (i.e. $f_1 = 0, f_2 > 1$ to $f_1 = 0, f_2 = 1$, and $f_1 > 1, f_2 = 0$ to $f_1 = 1, f_2 = 0$). Remarkably, among the 512 initial points $\mathbf{w}_0$ considered for grid search within the $[-1, 1]^3$ space, 372 of them lead to inferior stationary solutions, accounting for 72.66% of the cases. This observation suggests that MGDA has a notable tendency to converge to suboptimal solutions even when starting from a uniformly distributed set of initial weights.

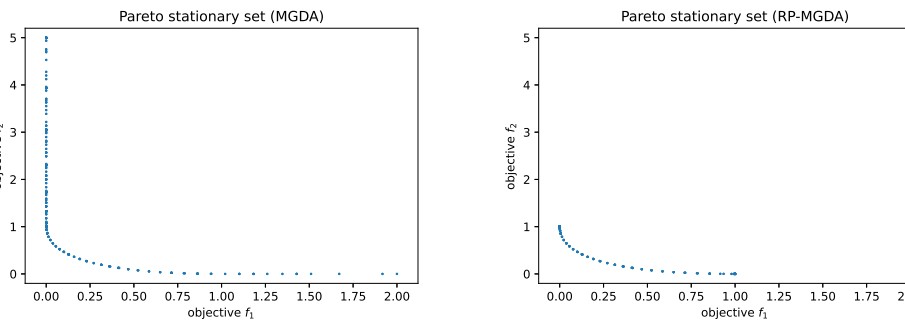

Figure 4: Comparison of Pareto stationary points generated by MGDA and RPMGDA on individually strictly convex objectives. We can see clearly that MGDA (**Left**) can converge to sub-optimal Pareto stationary solutions, while RP-MGDA (**Right**) always reaches solutions on the Pareto front.

### 5.1.1 DOES ADDING REGULARIZATION TO INDUCE STRONG CONVEXITY HELP?

We have discussed in §3.2 on the importance of joint strict convexity for the performance of Pareto stationary solutions. One natural attempt is to add regularization to the objectives to induce strong convexity. We run MGDA on the regularized objectives with different degrees of regularization.

We observe in Figure 5 that bigger regularization yields solutions that are further away from Pareto front; while smaller regularization gradually recover the previous (undesired) Pareto stationary set of MGDA (as in Figure 4), thus having more inferior weakly Pareto solutions (i.e. those on the tails). To summarize, adding regularization term (big or small) to the objective to induce strong convexity does *not* help alleviate the issue of MGDA, which further suggests that the drawback may only be solved by taking the variable structure into account.

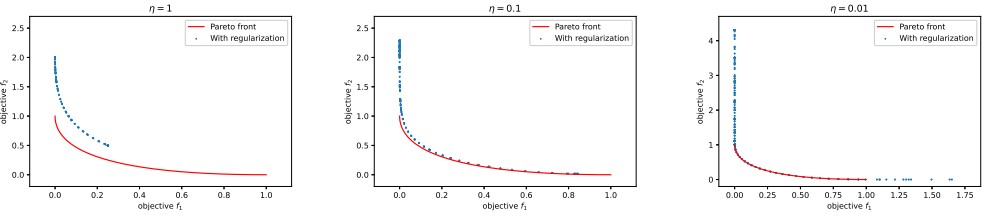

Figure 5: Frontier of MGDA solutions from regularized objectives, and comparison with the true Pareto front (red). Regularization factor $\eta$ from big to small.

### 5.1.2 NON STRICTLY CONVEX OBJECTIVE

Here we also investigate the case where $f_1$ and $f_2$ are not strictly convex w.r.t. the variables they depend on. Consider $f_1(\boldsymbol{\theta}) = \theta_1^2 + |\theta_2|$, $f_2(\boldsymbol{\theta}) = |\theta_2 - 1| + (\theta_3 - 1)^2$ .We demonstrate in Figure 6 that MGDA fails in a similar way, while RP-MGDA still recovers the Pareto front.

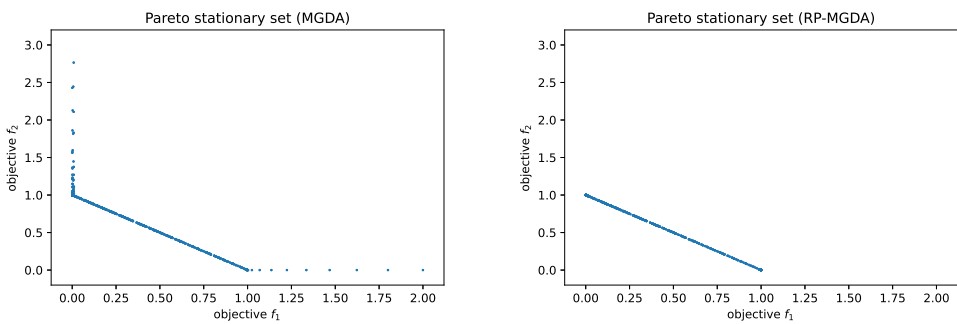

Figure 6: Comparison of Pareto stationary points generated by MGDA and RPMGDA on individually non-strictly convex objectives. Initializations are uniformly distributed and chosen with grid search.

### 5.2 RANDOM DEPENDENCY MATRIX (5-BY-5)

In this section, we consider the quadratic setup: $f_i = \sum_{j \in e(f_i)} (\theta_j - i)^2$, where $e(f_i)$ is the set of variable indices that $f_i$ depends on. The variable dependency structure is given in Figure 3.

We apply refined partitioning procedure to this problem: By rule (II), $(\theta_3, \theta_4, \theta_5)$ are merged since a loop with different colors of length 3 exists when examining $f_3, f_4, f_5$. Next, let $\boldsymbol{\theta}' = (\theta_3, \theta_4, \theta_5)$. By rule (III), the remaining variables $\theta_1, \theta_2, \boldsymbol{\theta}'$ are partitioned and optimized separately. So the resulting RP-MGDA is performing (6) w.r.t. $\boldsymbol{\theta}' = (\theta_3, \theta_4, \theta_5)$ and gradient descent w.r.t $\theta_1$ and $\theta_2$.

Results are shown in Figure 7. We see that the solutions of RP-MGDA dominates the solutions of MGDA, though all solutions are Pareto stationary (with corresponding dual coefficients plotted).

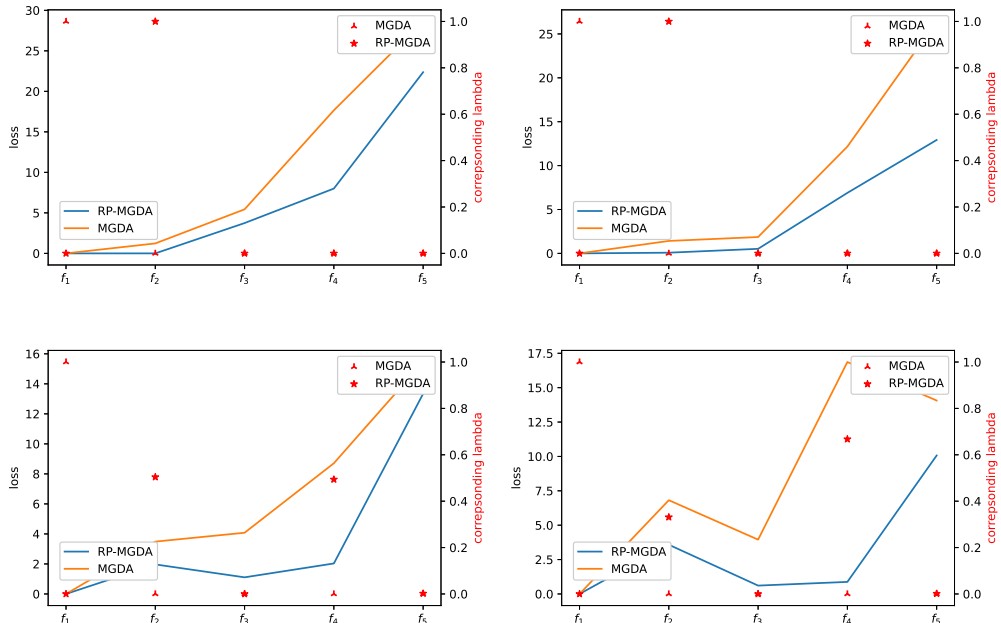

Figure 7: Comparison of MGDA and RP-MGDA solution quality, with random initializations. **(i)** Colored lines plot the final objective values of each $f_i$, where blue represents RP-MGDA, and orange represents MGDA. Clearly, the solutions of RP-MGDA Pareto dominates solutions of MGDA. **(ii)** We also plot, with red markers and right y-axis, the respective dual coefficients $\lambda_i$ of the final stationary solutions given by MGDA and RP-MGDA which validate the differences from a dual perspective.

## 6 CONCLUSIONS AND DISCUSSION

In this work, we identify a maybe overlooked drawback of MGDA algorithm applied to problems with specific variable dependency structures. Motivated from overcoming the drawback of MGDA in several representative scenarios (i.e. the toy example in §3.2 and personalized federated learning setting in §A.2), we propose RP-MGDA to improve the performance of MGDA by considering the variable structure and do refined partitioning instead of naive MGDA. We demonstrated theoretically that RP-MGDA is at least as good as MGDA, and beyond that, RP-MGDA outperforms MGDA in many cases. We give a detailed investigation of why MGDA fails in these cases and why the issue cannot be fixed by other intuitive attempts without considering the variable structure. Empirically, we demonstrate the effectiveness of RP-MGDA through variou examples. We believe our work is important to the field of multi-objective optimization, and hope it will be a useful reference for other (maybe future) MGDA-type methods.

We introduce a novel concept of *refined partitioning* that holds promise for enhancing multi-objective gradient methods other than MGDA. RP-MGDA represents an initial exploration of this concept, opening the door to a new paradigm of improving multi-objective optimization through variable partitioning. This innovative approach paves the way for future research into the broader applicability of refined partitioning techniques across a spectrum of multi-objective optimization algorithms, e.g. PCGrad (Yu et al., 2020) and CAGrad (Liu et al., 2021).

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
