# A APPENDIX

## A.1 PROOFS

In order to prove Theorem 1, we need to first look at a special variable structure, which is a 'chain'.

**A non-trivial case: Chain (no loop)**    A chain is a variable dependency structure illustrated below of shape $(n + 1)$ x $n$, (below is the case $n = 3$). It is clear that any more dependency entry added upon a chain will either result in a loop or a dense block.

For the chain example, we argue that we can actually perform min-norm oracle (6) separately w.r.t $\boldsymbol{\theta}_1, \boldsymbol{\theta}_2, \boldsymbol{\theta}_3$ and this guarantees Pareto stationary solutions upon termination. Although the proof is shown for the case $n = 3$ below, for simplicity, it is not hard to generalize to any $n \geq 3$.

We prove this by writing out the exact characterization for the stationary solutions upon termination and show it is strictly included in the set of all Pareto stationary solutions.

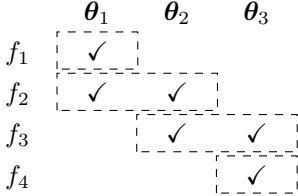

**Lemma 2.** *RP-MGDA applied to the chain structure terminates only upon Pareto stationary solutions.*

*Proof.* The RP-MGDA solutions $\mathbf{w}^* = (\boldsymbol{\theta}_1^*, \boldsymbol{\theta}_2^*, \boldsymbol{\theta}_3^*)$ in this example satisfy the following condition:

$$\begin{cases} \lambda_1' \nabla_{\boldsymbol{\theta}_1} f_1(\mathbf{w}^*) + \lambda_2' \nabla_{\boldsymbol{\theta}_1} f_2(\mathbf{w}^*) = \mathbf{0} \\ \lambda_2'' \nabla_{\boldsymbol{\theta}_2} f_2(\mathbf{w}^*) + \lambda_3'' \nabla_{\boldsymbol{\theta}_2} f_3(\mathbf{w}^*) = \mathbf{0} \\ \lambda_3' \nabla_{\boldsymbol{\theta}_3} f_3(\mathbf{w}^*) + \lambda_4' \nabla_{\boldsymbol{\theta}_3} f_4(\mathbf{w}^*) = \mathbf{0} \end{cases} \tag{9}$$

we can then find $(\lambda_1, \lambda_2, \lambda_3, \lambda_4)$ that satisfies the Pareto stationary condition

$$\lambda_1 \begin{bmatrix} \nabla_{\boldsymbol{\theta}_1} f_1(\mathbf{w}^*) \\ \nabla_{\boldsymbol{\theta}_2} f_1(\mathbf{w}^*) \\ \nabla_{\boldsymbol{\theta}_3} f_1(\mathbf{w}^*) \end{bmatrix} + \lambda_2 \begin{bmatrix} \nabla_{\boldsymbol{\theta}_1} f_2(\mathbf{w}^*) \\ \nabla_{\boldsymbol{\theta}_2} f_2(\mathbf{w}^*) \\ \nabla_{\boldsymbol{\theta}_3} f_2(\mathbf{w}^*) \end{bmatrix} + \lambda_3 \begin{bmatrix} \nabla_{\boldsymbol{\theta}_1} f_3(\mathbf{w}^*) \\ \nabla_{\boldsymbol{\theta}_2} f_3(\mathbf{w}^*) \\ \nabla_{\boldsymbol{\theta}_3} f_3(\mathbf{w}^*) \end{bmatrix} + \lambda_4 \begin{bmatrix} \nabla_{\boldsymbol{\theta}_1} f_4(\mathbf{w}^*) \\ \nabla_{\boldsymbol{\theta}_2} f_4(\mathbf{w}^*) \\ \nabla_{\boldsymbol{\theta}_3} f_4(\mathbf{w}^*) \end{bmatrix} = \mathbf{0} \tag{10}$$

by keeping ratios the same. Specifically, choose $(\lambda_1, \lambda_2, \lambda_3, \lambda_4)$ such that $\lambda_1 : \lambda_2 = \lambda_1' : \lambda_2'$, $\lambda_2 : \lambda_3 = \lambda_2'' : \lambda_3''$, and $\lambda_3 : \lambda_4 = \lambda_3' : \lambda_4'$. □

**Theorem 1.** RP-MGDA must attain a Pareto stationary solution upon termination.

*Proof.* We can show that after exhaustively applying rule (I) and (II), the final variable dependency structure we arrive at, just before applying rule (III), is either a chain or sparser than a chain (A is sparser than B if by adding more dependency entries to A it can become B). Notice that if the shape does not match (e.g., many objectives and few variables), we can always add dummy variables.

The resulting dependency structure cannot be denser than a chain since any addition of an entry must result in either a loop or a dense block, which contradicts rule (I) or (II).

Now that we have shown already that RP-MGDA applied to a chain guarantees Pareto stationary solutions, and since sparser structures only exhibit fewer constraints over $\lambda's$ when reaching stationary solutions, applying RP-MGDA to the final variable dependency structure guarantees Pareto stationarity. □

## A.2 SCENARIO: PERSONALIZED FEDERATED LEARNING

Let's consider a personalized federated learning setting where every client share a global variable (i.e., model parameters) $\mathbf{x}$, and each client also owns a local preference variable (e.g. a local model that

extracts compact features) $\mathbf{z}_i$, so that each objective function $f_i$ is specified by two variables $(\mathbf{x}, \mathbf{z}_i)$ and thusf can be written as $f_i(\mathbf{x}, \mathbf{z}_i)$. If we apply MGDA naively, we will treat $\mathbf{w} = (\mathbf{x}, \mathbf{z}_1, \ldots, \mathbf{z}_n)$ while neglecting the sparse variable structure.

The variable structure can be understood more clearly with graphical illustration. For example when $n = 3$, we can draw it as:

$$
\begin{array}{c c c c c}
 & \mathbf{x} & \mathbf{z}_1 & \mathbf{z}_2 & \mathbf{z}_3 \\
f_1 & \checkmark & \checkmark & & \\
f_2 & \checkmark & & \checkmark & \\
f_3 & \checkmark & & & \checkmark
\end{array}
$$

As we have mentioned, a naive way to apply MGDA is to treat $\mathbf{w} = (\mathbf{x}, \mathbf{z}_1, \mathbf{z}_2, \mathbf{z}_3)$ and proceeds as (6). However, similar to the idea in 3.2, we can do much better by first partitioning according to variables and optimize in a somewhat coordinate-wise way.

By applying RP-MGDA, we are in effect applying MGDA to the $\mathbf{x}$ coordinate of $f_1, f_2, f_3$ and apply separate gradient descent to the $\mathbf{z}_i$ coordinate of $f_i$. The refined optimization iteration is:

$$
\mathbf{x}^{t+1} = \mathbf{x}^t - \eta \mathbf{d}^t, \quad \mathbf{d}^t = J_{\mathbf{f}}(\mathbf{x}^t)\boldsymbol{\lambda}_*^t,
$$
$$
\text{where} \quad \boldsymbol{\lambda}_*^t = \operatorname*{argmin}_{\boldsymbol{\lambda} \in \Delta} \|J_{\mathbf{f}}(\mathbf{x}^t)\boldsymbol{\lambda}\|^2. \tag{11}
$$
$$
\mathbf{z}_i^{t+1} = \mathbf{z}_i^t - \eta \nabla_{\mathbf{z}_i} f_i(\mathbf{x}^t, \mathbf{z}_i^t), \tag{12}
$$

We will show that this refined MGDA update rule (RP-MGDA) reaches better solutions:

1. In Proposition 1, we prove that the set of solutions of RP-MGDA is a strict subset of (and thus superior to) Pareto stationary solutions, i.e. solutions of naive MGDA.

2. In Proposition 2 we prove that, under reasonable convexity assumptions, the solutions of this refined approach (12) are Pareto optimal while the ones of naive MGDA are not.

**Proposition 1.** *All solutions of Algorithm* (12) *are Pareto stationary, but not vice versa.*

*Proof.* The stationary solutions of 12 are characterized by

$$
\nabla_{\mathbf{z}_i} f_i(\mathbf{x}^*, \mathbf{z}_i^*) = 0, \forall i, \quad \text{and}
$$
$$
\exists \lambda_i \geq 0, \sum_{i=1}^{n} \lambda_i = 1, \quad s.t.
$$
$$
\sum_{i=1}^{n} \lambda_i \nabla_{\mathbf{x}} f_i(\mathbf{x}^*, \mathbf{z}_i^*) = 0 \tag{13}
$$

This implies that

$$
0 = \sum_{i=1}^{n} \lambda_i \begin{bmatrix} \nabla_{\mathbf{x}} f_i(\mathbf{x}^*, \mathbf{z}_i^*) \\ \vdots \\ \nabla_{\mathbf{z}_i} f_i(\mathbf{x}^*, \mathbf{z}_i^*) \\ \vdots \end{bmatrix} = \sum_{i=1}^{n} \lambda_i \nabla f_i(\mathbf{x}^*, \mathbf{z}^*) \tag{14}
$$

which is exactly the definition of Pareto stationarity.

On the other hand, the Pareto stationary condition $\sum_{i=1}^{n} \lambda_i \nabla f_i(\mathbf{x}^*, \mathbf{z}^*) = 0$ doesn't necessarily imply (13), since $\nabla_{\mathbf{z}_i} f_i(\mathbf{x}^*, \mathbf{z}_i^*)$ can be nonzero for indices $i$ where $\lambda_i = 0$. $\qquad\square$

**Lemma 3.** *If $f(\mathbf{x}, \mathbf{y})$ is strictly convex w.r.t $(\mathbf{x}, \mathbf{y})$, and $g(\mathbf{x}, \mathbf{z})$ is strictly convex w.r.t $(\mathbf{x}, \mathbf{z})$. Then $f + g$ is strictly convex w.r.t. $(\mathbf{x}, \mathbf{y}, \mathbf{z})$.*

*Proof.* By definition, $f + g$ is strictly convex w.r.t $(\mathbf{x}, \mathbf{y}, \mathbf{z})$ iff $\forall (\mathbf{x}_1, \mathbf{y}_1, \mathbf{z}_1) \neq (\mathbf{x}_2, \mathbf{y}_2, \mathbf{z}_2)$ and $0 < t < 1$ (let $\mathbf{w}_1 := (\mathbf{x}_1, \mathbf{y}_1, \mathbf{z}_1)$ and $\mathbf{w}_2 := (\mathbf{x}_2, \mathbf{y}_2, \mathbf{z}_2)$), we have

$$
(f + g)(t\mathbf{w}_1 + (1 - t)\mathbf{w}_2) < t(f + g)(\mathbf{w}_1) + (1 - t)(f + g)(\mathbf{w}_2) \tag{15}
$$

Note that

$$
\begin{aligned}
\text{LHS} &= f(t(\mathbf{x}_1, \mathbf{y}_1) + (1-t)(\mathbf{x}_2, \mathbf{y}_2)) + g(t(\mathbf{x}_1, \mathbf{z}_1) + (1-t)(\mathbf{x}_2, \mathbf{z}_2)) \\
&< tf(\mathbf{x}_1, \mathbf{y}_1) + (1-t)f(\mathbf{x}_2, \mathbf{y}_2) + tg(\mathbf{x}_1, \mathbf{z}_1) + (1-t)g(\mathbf{x}_2, \mathbf{z}_2) \\
&= t(f+g)(\mathbf{x}_1, \mathbf{y}_1, \mathbf{z}_1) + (1-t)(f+g)(\mathbf{x}_2, \mathbf{y}_2, \mathbf{z}_2)
\end{aligned}
\tag{16}
$$

where the first inequality is due to strict convexity of $f(\mathbf{x}, \mathbf{y})$ and $g(\mathbf{x}, \mathbf{z})$. Q.E.D. $\qquad\square$

**Proposition 2.** *With variable structure in the above PFL setting, and jointly convex $f_i$ (but not necessary strictly), the solutions of* (12) *are Pareto optimal, given that each $f_i$ is strictly convex w.r.t.* $(\mathbf{x}, \mathbf{z}_i)$.

*Proof.* Notation, let $\mathbf{w}^* = (\mathbf{x}^*, \mathbf{z}_1^*, \mathbf{z}_2^*, \mathbf{z}_3^*)$ be the stationary solution of Alg 12, let $\mathbf{w}' = (\mathbf{x}', \mathbf{z}_1', \mathbf{z}_2', \mathbf{z}_3')$ be a potential candidate solution that Pareto dominates $\mathbf{w}^*$. In shorthand, denote $f_i^* := f_i^*(\mathbf{x}^*, \mathbf{z}_i^*), f_i' := f_i'(\mathbf{x}', \mathbf{z}_i')$.
Recall the stationarity condition of 12:

$$
\nabla_{\mathbf{z}_i} f_i(\mathbf{x}^*, \mathbf{z}_i^*) = 0, \forall i, \text{ and}
$$
$$
\exists \lambda_i \geq 0, \sum_{i=1}^{n} \lambda_i = 1, \ s.t.
\tag{17}
$$
$$
\sum_{i=1}^{n} \lambda_i \nabla_{\mathbf{x}} f_i(\mathbf{x}^*, \mathbf{z}_i^*) = 0
$$

If all $\lambda_i$ are nonzero, then $\mathbf{w}^*$ is Pareto optimal, since by Proposition 1, $\mathbf{w}^*$ is always Pareto stationary, and Pareto stationary solutions are (properly) Pareto optimal if all $\lambda_i$ are nonzero (see Jahn's book, it is quite easy to give a separate proof, and properly Pareto optimal is actually stronger). Q.E.D.
If one of the $\lambda_i$ is 0, WLOG, say $\lambda_1 = 0, \lambda_2, \lambda_3 \neq 0$. Consider an auxiliary function $\bar{f}(\mathbf{x}, \mathbf{z}_2, \mathbf{z}_3) := \lambda_2 f_2 + \lambda_3 f_3$, since $\bar{f}$ is strictly convex (by Lemma 3), $\nabla \bar{f}(\cdot) = 0$ has only one unique solution and is the minimizer. Furthermore,

$$
\nabla \bar{f}(\mathbf{x}^*, \mathbf{z}_2^*, \mathbf{z}_3^*) = \lambda_2 \begin{bmatrix} \nabla_{\mathbf{x}} f_2(\mathbf{x}^*, \mathbf{z}_2^*) \\ \nabla_{\mathbf{z}_2} f_2(\mathbf{x}^*, \mathbf{z}_2^*) \\ 0 \end{bmatrix} + \lambda_3 \begin{bmatrix} \nabla_{\mathbf{x}} f_3(\mathbf{x}^*, \mathbf{z}_3^*) \\ 0 \\ \nabla_{\mathbf{z}_3} f_3(\mathbf{x}^*, \mathbf{z}_3^*) \end{bmatrix} = 0
$$

So $(\mathbf{x}^*, \mathbf{z}_2^*, \mathbf{z}_3^*)$ is the unique minimizer of $\bar{f}$, and thus if $\mathbf{w}'$ dominates $\mathbf{w}^*$, we must have $(\mathbf{x}', \mathbf{z}_2', \mathbf{z}_3') = (\mathbf{x}^*, \mathbf{z}_2^*, \mathbf{z}_3^*)$.
Now that $\mathbf{w}'$ dominates $\mathbf{w}^*$, it must follow that $f_1' < f_1^*$. However, that means $f_1(\mathbf{x}^*, \mathbf{z}_1') < f_1(\mathbf{x}^*, \mathbf{z}_1^*)$, which is not possible because $f_1(\mathbf{x}^*, \cdot)$ is convex and $\nabla_{\mathbf{z}_1} f_1(\mathbf{x}^*, \mathbf{z}_1^*) = 0$.
Q.E.D. The same proof applies to arbitrary $n$ and arbitrary number of zeros $\lambda_i$. $\qquad\square$

**Remark 4** (Feasibility of assumption). *Note that in Proposition 2, with the PFL variable structure, it is impossible for any $f_i$ to be strictly convex (jointly). Thus, naive MGDA can **never** guarantee Pareto optimality for this problem. On the other hand, assumptions on strict convexity w.r.t. partial variables $(\mathbf{x}, \mathbf{z}_i)$ is feasible, e.g. see Example below.*

With the scenario described in §A.2, Theorem 2 is proved with Proposition 2 and Remark 4.

We also empirically justify the above claims through an example.

**A personalized federated learning example**. We consider the PFL setting with $n = 3$ clients. $f_1(x, z_1, z_2, z_3) = x^2 + z_1^2$, $f_2(x, z_1, z_2, z_3) = (x-1)^2 + z_2^2$, $f_3(x, z_1, z_2, z_3) = (x-2)^2 + z_3^2$. According to experiments, with step size 0.01 and 1000 iterations. Apply MGDA with initialization $(x, z_1, z_2, z_3) = (1.5, 1, 0.1, 0.1)$, the algorithm converges to $(x, z_1, z_2, z_3) = (1.5, 1, 0, 0)$ with function values $(f_1, f_2, f_3) = (\frac{13}{4}, \frac{1}{4}, \frac{1}{4})$. This is not Pareto optimal since it is dominated by $(f_1, f_2, f_3) = (\frac{9}{4}, \frac{1}{4}, \frac{1}{4})$ when $(x, z_1, z_2, z_3) = (1.5, 0, 0, 0)$. In contrast, the refined approach 12, with same setting, converges to $(x, z_1, z_2, z_3) = (1.5, 0, 0, 0)$ which is Pareto optimal.

From Proposition 1, we also see why MGDA fails to reach Pareto optimal solution in the previous example (since in the example, $\lambda_1 = 0$, $\nabla_{z_1} f_i(x^*, z_1^*) \neq 0$, and MGDA stops even though it has a descent direction along $z_1$).