# OpenReview forum: "Refined Partitioning Boosts MGDA: Introducing RP-MGDA for Multi-Objective Learning"
_ICLR.cc/2024/Conference — Submitted to ICLR 2024_

### Official Review · Reviewer_wLSR · 2023-10-30

**Soundness:** 2 fair
**Presentation:** 2 fair
**Contribution:** 2 fair
**Rating:** 5
**Confidence:** 3

**Summary:**

This paper studies the weak optimality issue of MGDA in multi-objective optimization and proposes a new algorithm based on the refined partitioning technique to escape the weakly Pareto optimal points. Specifically, this paper first provides a concrete example, showing that the optimization process of vanilla MGDA may stuck in weakly Pareto points. Enlightened by this observation, this paper proposes to partition the model parameters into different groups according to three partitioning rules, which are then fed to the min-norm solver separately. A case study on personalized federated learning shows the effectiveness of the RP-MGDA's achieving strict Pareto solutions, which is also supported by empirical results.

**Strengths:**

1. The analysis on weakly Pareto optimal issue for MGDA and the algorithm partitioning all parameters is interesting. The proposed three partitioning rules sound non-trivial.

2. The case study on personalized federated learning verifies the effectiveness of the proposed algorithm in specific scenarios.

3. This paper is generally well written and easy to follow.

**Weaknesses:**

1. Beyond the specific showcases, it would be better to provide more explanations on the weak optimality issue for MGDA and partitioning everything in principle. Moreover, the intuition of the proposed partitioning rules is not properly discussed. It is not clear whether there is any design principle for partitioning, and more specifically, how do these rules help the MOO algorithm escape the weak Pareto optimal points.

2. Although the case study on personalized federated learning is interesting, a discussion on the effectiveness of RP-MGDA in more general scenarios will make this paper more significant and technically sound.

**Questions:**

1. It was claimed that the proposed partitioning technique can also be applied to other multi-objective gradient methods such as PCGrad and CAGrad. Can you justify this point?

2. I would recommend the author to give a more thorough literature review on the related work section.

---

### Official Review · Reviewer_pzQ4 · 2023-10-31

**Soundness:** 2 fair
**Presentation:** 2 fair
**Contribution:** 2 fair
**Rating:** 3
**Confidence:** 4

**Summary:**

This paper proposed a refined partition based multiple gradient descent algorithm (RP-MGDA) for multi-objective optimization. It first identifies and analyzes a potential drawback for the widely-used MGDA, where MGDA might only find Pareto stationary but not Pareto optimal solution when all the objective functions are only strictly convex for partial variables but not for all variables. Based on this observation, this work proposes a refined partition approach that divides the variables $x$ into different groups based on the variable structure, and then separately optimizes them with MGDA during the optimization process. The experiment shows that RP-MGDA can find Pareto optimal solutions for some synthetic toy problems while the original MGDA cannot.

**Strengths:**

+ This work is well-organized and easy to follow.

+ Multi-objective optimization can be found in many real-world applications, where the multiple gradient descent algorithm (MGDA) is a widely-used algorithm. The proposed method can potentially address a drawback for MGDA but with several major concerns (see weaknesses).

**Weaknesses:**

**1. Assumption on Completely Known Variable Structure**

The original MGDA only depends on the standard first-order oracle for multi-objective optimization, where it can query a variable $x$ and then receive $(f_i(x), \nabla f_i(x))$ pairs for all objective functions. The variable structure is a black-box for MGDA. In contrast, the proposed method has a much stronger assumption that all the variable relations among $x$ for all objective functions are completely known. This important difference is not properly discussed in the paper.

**2. Novelty over the Current Practice**

Although the original MGDA works on the whole variables, a simple **separately optimizing the variables specific to each objective, and then tackling the rest variables shared by more than one objective with MGDA** approach is already widely used in practice. For example, see the seminal work [1] and many follow-ups in the recent years.

The proposed RP-MGDA has two rules to merge the variables, namely, the 'dense block' merge and the 'loop' merge. However, it seems that all the toy examples (section 5.1 and section 5.2) as well as the personalized federated learning example (section A.2) all just simply optimize the objective-specific variable separately, and then optimize the rest variables together.

What is the novelty and advantage of the proposed RP approach over the current practical approach? Can you give an example that how the RP-MGDA will lead to a different grouping? Is there any real-world application to demonstrate and support this setting?

**3. Experiments on Real-World Applications**

This work only conducts experiments on a few toy problems, and the personalized federated learning example (in the Appendix) is an illustrative prototype without experiment.

As a methodology paper, I expect it should have comprehensive experiments on real-world applications that the audience of the ICLR community will be interested in, like those in [1-5]. The current theoretical papers usually also have experiments on large-scale real-world applications like deep multi-task learning [6, 7].

**Questions:**

- Please address the concerns raised in the above weaknesses.

- The illustrative example (equation 8 on page 4) is not appropriate. Once the variable structure is known, it immediately reduces to two independent and unconflicted single-objective problems.

- Please make sure the citation (\citet, \citep) is properly used in the paper.

######################## Post-Rebuttal Comment #########################

I've read other reviewers' comments and the general response from authors, and decide to keep my current score.

The variable structure is an important research issue for multi-objective optimization. I encourage the author to further investigate and improve the proposed Refined Partitioning method for developing an efficient multi-objective optimization algorithm.

**Reference**

[1] Multi-task learning as multi-objective optimization. NeurIPS 2018.

[2] Gradient surgery for multi-task learning. NeurIPS 2020.

[3] Conflict-averse gradient descent for multi-task learning. NeurIP 2020.

[4] Impartial Multi-Task Learning. ICLR 2021.

[5] Multi-Task Learning as a Bargaining Game. ICML 2022.

[6] Mitigating Gradient Bias in Multi-objective Learning: A Provably Convergent Approach. ICLR 2023.

[7] Three-Way Trade-Off in Multi-Objective Learning: Optimization, Generalization and Conflict-Avoidance. NeurIPS 2023.

---

### Official Review · Reviewer_kzPt · 2023-11-07

**Soundness:** 3 good
**Presentation:** 3 good
**Contribution:** 3 good
**Rating:** 5
**Confidence:** 4

**Summary:**

The authors found  the drawback of MGDA algorithm applied to problems with specific variable dependency structures. Motivated from this finding, they propose RP-MGDA to improve the performance of MGDA by considering the variable structure and do refined partitioning instead of naive MGDA. Theoretically, it was demonstrated that RP-MGDA is at least as good as MGDA. Empirically, they demonstrate the effectiveness of RP-MGDA through two toy examples.

**Strengths:**

1. I like it very much that the authors illustrate their ideas with toy examples throughout the paper. It really helps the understanding of the paper a lot.

2. To the best of my knowledge, the problem setting of MOO problems with general variable dependency structures is rarely (probably none) studied. Thus, I think it is quite interesting to see the analysis the authors show in this paper the impact of the findings.

**Weaknesses:**

I think the paper can be improved in the following ways:

1. More discussions about more related works of MGDA variants, such as on ICML paper https://proceedings.mlr.press/v162/momma22a/momma22a.pdf

Also,  for applications like federated MOO, the authors could also discuss about https://arxiv.org/pdf/2310.09866.pdf

2. The experiments studied in this paper is too small: one bi-objective non-convex optimization problem, and one RANDOM DEPENDENCY MATRIX (5-BY-5). In fact, based on my experience, algorithms that worked well in toy examples might not work at all or that well in real world examples. For example, https://dl.acm.org/doi/pdf/10.1145/3580305.3599870 reported that one really has too modified the proposed MGDA-based algorithms a bit to make it work.

Besides, it is virtually impossible to apply the proposed Refined Partitioning rules to real world ML datasets, such as those vision or language datasets.

**Questions:**

Although I like it very much about the refined partitioning ideas, can the authors help me understand how to apply the proposed Refined Partitioning rules to real world ML datasets, such as those vision or language datasets?

---

### Official Review · Reviewer_M8VM · 2023-11-11

**Soundness:** 3 good
**Presentation:** 3 good
**Contribution:** 3 good
**Rating:** 3
**Confidence:** 2

**Summary:**

This paper studied the performance of the widely used multi-gradient descent algorithm (MGDA) for multi-objective optimization (MOO). The authors constructed pedagogical examples to illustrate the sub-optimal performance of MGDA in terms of Pareto-optimality when the objective functions are convex. To address this problem and motivated by the examples, the authors proposed a refined partitioning MGDA (RP-MGDA) approach. The authors empirically showed that RP-MGDA achieves better solutions compared to the basic MGDA approach.

**Strengths:**

1. MOO is a timely topic in the learning community but remains relatively under-explored. This paper provides interesting insights and improves the understanding of the basic MGDA approach.

2. The proposed RP-MGDA approach is interesting and overcomes some limitations of MGDA.

**Weaknesses:**

1. This paper lacks theoretical depth and is mostly an empirical paper.

2. Some of the claims are questionable.

Please see the questions below.

**Questions:**

1. The constructed examples in this paper are mostly pedagogical toy examples or synthetic problems. Many of the examples are for convex objective functions. In practice, however, most problems are with non-convex objective functions. Thus, it remains clear whether the advantage of RP-MGDA over MGDA continues to hold under non-convex cases. Could the authors provide more examples for non-convex cases?

2. Theorem 2 seems odd and not very meaningful. It says that "there exists a convex multi-objective optimization problem where the solution of RM-MGDA are provably Pareto optimal where MGDA is not." However, based on simple logic, one cannot draw the conclusion that RP-MGDA is better than MGDA (even for the convex case), right?

---

### Author Response · Authors · 2023-11-22

We thank all the reviewers for their time and effort to give insightful and constructive comments to this paper. We are glad that reviewers find our paper easy to follow and our proposed approach being interesting. We appreciate reviewer kzPt and wLSR's recognition that MOO problems with general variable dependency structures is rarely studied and our refined partitioning idea is novel and non-trivial. We are happy to include more comprehensive discussions in the related works section with reference to the suggested papers and more.

We acknowledge reviewer's concern about the applicability of RP-MGDA to more complex empirical problems, for example, non-convex optimization and more specifically real world machine learning problems.  We very much understand the concern and will work on experiments with CNN models on image classification tasks as our next step. We also agree that examples of more real world scenarios (besides the personalized federated learning setting) will make this paper more significant. At this point, we are aware of some other special variable dependency structures for real world machine learning problems, for example, some autoregressive neural network structures ([1][2]) have a lower-triangular dependency matrix. We will include these discussions in the future version of this manuscript.

Finally, we would like to restate that the refined partitioning rules we come up with are non-trivial (backed up by the finding in Section 4.1 and proof for Theorem 1), and that RP-MGDA is a generalization of MGDA which in the worst case it just reduces to MGDA while in cases where a finer partition can be found, it is better to use RP-MGDA. The superiority is both (i) empirically verified and (ii) can be theoretically established------where in the most general scenario it enjoys Pareto stationarity guarantee as MGDA does ("at least as good"), and in more specific cases it can guarantee Pareto optimality while MGDA cannot, even under the same convexity assumptions ("can be better than"). We thank the reviewers for this suggestion aimed at making our work more clear.

[1] Sum-of-Squares Polynomial Flow. ICML 2019
[2] DeeBERT: Dynamic Early Exiting for Accelerating BERT Inference. ACL 2020

---

### Meta-Review · Area_Chair_ZWN5 · 2023-12-05

**Metareview:**

This paper studied the multi-gradient descent algorithm for multi-objective optimization and addresses the challenge of sub-optimal Pareto solutions when objective functions are convex based on the idea of refined partition.

The reviewers' agree that the paper needs better discussion of the related work to contextualize the paper's contributions and improve the empirical evaluation on realistic benchmarks. Therefore, I recommend rejecting the paper and encourage the authors' to revise the paper based on the review comments for re-submission.

**Justification For Why Not Higher Score:**

Significant weaknesses as mentioned in the meta review.

**Justification For Why Not Lower Score:**

N/A

---

### Decision · Program_Chairs · 2024-01-16

Reject